# Genotypic and Phenotypic Detection of Polyhydroxyalkanoate Production in Bacterial Isolates from Food

**DOI:** 10.3390/ijms24021250

**Published:** 2023-01-08

**Authors:** Daniela Máčalová, Magda Janalíková, Jana Sedlaříková, Iveta Rektoříková, Marek Koutný, Pavel Pleva

**Affiliations:** 1Department of Environmental Protection Engineering, Faculty of Technology, Tomas Bata University in Zlin, 275 Vavreckova, 76001 Zlin, Czech Republic; 2Department of Fat, Surfactant and Cosmetics Technology, Faculty of Technology, Tomas Bata University in Zlin, 275 Vavreckova, 76001 Zlin, Czech Republic

**Keywords:** biomaterial, bacterial strains, biosynthetic pathways, polyhydroxyalkanoate, screening

## Abstract

Polyhydroxyalkanoates (PHAs) are widely used in medical and potentially in other applications due to their biocompatibility and biodegradability. Understanding PHA biosynthetic pathways may lead to the detection of appropriate conditions (substrates) for producing a particular PHA type by a specific microbial strain. The aim of this study was to establish a method enabling potentially interesting PHA bacterial producers to be found. In the study, all four classes of PHA synthases and other genes involved in PHA formation (*fabG*, *phaA*, *phaB*, *phaG*, and *phaJ*) were detected by PCR in 64 bacterial collection strains and food isolates. *Acinetobacter*, *Bacillus*, *Cupriavidus*, *Escherichia*, *Klebsiella*, *Lelliottia*, *Lysinibacillus*, *Mammaliicoccus*, *Oceanobacillus*, *Pantoea*, *Peribacillus*, *Priestia*, *Pseudomonas*, *Rahnella*, *Staphylococcus*, and *Stenotrophomonas* genera were found among these strains. Fructose, glucose, sunflower oil, and propionic acid were utilized as carbon sources and PHA production was detected by Sudan black staining, Nile blue staining, and FTIR methods. The class I synthase and *phaA* genes were the most frequently found, indicating the strains’ ability to synthesize PHA from carbohydrates. Among the tested bacterial strains, the *Pseudomonas* genus was identified as able to utilize all tested carbon sources. The *Pseudomonas extremorientalis* strain was determined as a prospect for biotechnology applications.

## 1. Introduction

Polyhydroxyalkanoates (PHAs) are a group of biodegradable, biocompatible polyesters that some microorganisms can synthesize in inclusions as energy storage molecules. Based on the number of carbons, PHAs are classified as short-chain-length (scl) PHAs with 3–5 carbons, medium-chain-length (mcl) PHAs with 6–14 carbons, and long-chain-length (lcl) PHAs with more than 15 carbons [1]. Due to these monomer composition variations, these polymers can have various properties, e.g., mechanical characteristics. Scl PHAs are inelastic and fragile polymers, which is due to their higher crystallinity. The elasticity of PHAs increase with the number of carbons in the monomers. To obtain ideal properties for a given application, copolymers with different monomers can be formed. Due to their biodegradability, PHAs have the potential for use as a packaging material and in the production of agricultural films [2,3]. It is likely that over time, PHAs will even partially replace typical packaging materials such as polyethylene, polypropylene, and polyethylene terephthalate [4]. Another advantage is the biocompatibility of PHAs with blood and tissues [5]. As a result, PHAs can be used in a wide range of commercial and biomedical applications [6,7]. However, the cost of PHA is still substantially higher than that of commonly used polymers for common applications such as packaging, so there is an effort to reduce the cost of these materials [8]. Options for the price reduction include using waste materials and searching for new, more capable microbial producers [9]. Understanding biosynthetic pathways can also help to reduce the cost and to optimize application-specific PHAs [10].

Several biosynthetic pathways of PHA have been described [11]. Only the pathways investigated here are described in the following text and Figure 1. The resulting metabolism and the formation of a given type of PHA depends on the nutritional status of the given microorganism [12]. 

PHAs can be synthetized from related substrates that serve as monomer precursors. The resulting PHA then, in general, corresponds to the original substrate. Most of these precursors are different variants of fatty acids processed by PHA producers into PHA monomers using the β-oxidation enzymatic pathway. This PHA synthesis pathway is important for producing mcl PHAs [13]. Intermediate products of β-oxidation (enoyl-CoA and 3-ketoacyl-CoA) can be used to form PHA by other enzymes. In branch A (Figure 1), 3-ketoacyl-CoA is used to produce PHA via 3-ketoacyl reductase (FabG), forming R-3-hydroxyacyl-CoA PHA monomers [14]. Pathway B (Figure 1) shows the use of enoyl-CoA by the R-specific enoyl hydratase (phaJ) to form PHA monomers [15].

**Figure 1 ijms-24-01250-f001:**
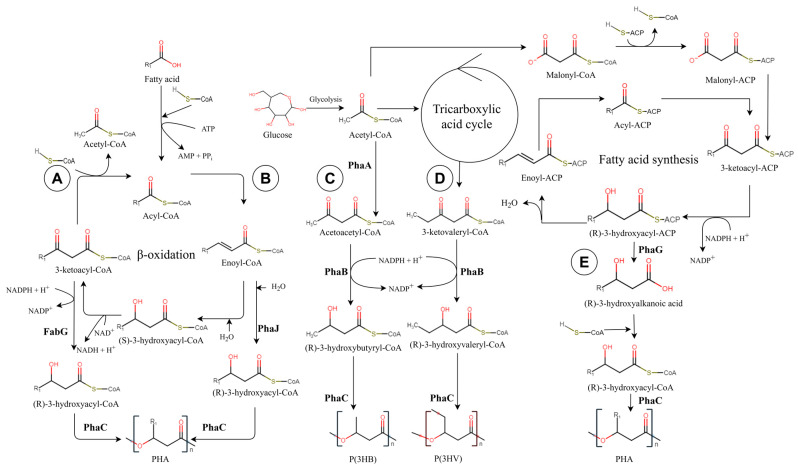
PHA biosynthetic pathways: (**A**,**B**) Production of PHAs from related carbon sources via β-oxidation; (**C**,**D**) production of scl PHAs from unrelated carbon sources; (**E**) production of mcl PHAs via fatty acid synthesis. (FabG: 3-ketoacyl reductase; PhaA: 3-ketothialase; PhaB: NADPH acetoacetyl-CoA reductase; PhaC: PHA synthase; PhaG: hydroxyacyl-ACP specific thioesterase; PhaJ: R-specific enoyl hydratase; PHA: polyhydroxyalkanoate; P(3HB): poly(3-hydroxybutyrate); and P(3HV): poly(3-hydroxyvalerate)). (Adapted from [14,15,16,17,18]).

Carbohydrates, i.e., glucose, are another substrate that can produce scl PHAs. At the beginning of the pathway, glucose is metabolized in glycolysis to form pyruvate. During the aerobic growth, pyruvate is converted to acetyl-CoA. In pathway C (Figure 1), two molecules of acetyl-CoA are then condensed by 3-ketothialase (PhaA) to form acetoacetyl-CoA, which is subsequently reduced by NADPH acetoacetyl-CoA reductase (PhaB) to form hydroxybutyryl-CoA. The final step is the polymerization of hydroxybutyryl-CoA by PHA synthase (PhaC) to form poly(3-hydroxybutyrate) (P(3HB)) [16]. Pathway D (Figure 1) shows a way to produce poly(3-hydroxyvalerate) (P(3HV)) from succinyl-CoA formed in the tricarboxylic acid cycle. Succinyl-CoA is converted to 2-(R)-methylmalonyl-CoA by coenzyme B12-dependent methylmalonyl CoA mutase (Sbm), which is subsequently converted to propionyl-CoA by methylmalonyl-CoA decarboxylase (YgfG). Propionyl-CoA is then converted to 3-ketovaleryl-CoA and subsequently to (R)-3-hydroxyvaleryl-CoA by PhaB, which subsequently produces P(3HV) [17].

There are several pathways for mcl PHA biosynthesis from various carbon sources based on fatty acid synthesis. Unlike β-oxidation, which shortens fatty acyl substrates by two carbons to release acetyl-CoA in each cycle and where all intermediates are linked to CoA, fatty acid synthesis elongates the molecules by two carbons per cycle via intermediates linked to acyl carrier protein (ACP). Pathway E (Figure 1) utilizes the intermediate (R)-3-hydroxyacyl-ACP, which is transformed by hydroxyacyl-ACP specific thioesterase (PhaG) to release (R)-3-hydroxyalkanoic acid. This hydroxy acid can be activated by acyl-CoA synthase (AlkK) to form (R)-3-hydroxyacyl-CoA, which is subsequently used in PHA polymerization [18].

For the formation of PHA, a necessary enzyme in all mentioned pathways is PHA synthase, which polymerizes the monomer units and releases CoA. PHA synthases are divided into four classes, distinguished by their primary sequences, substrate specificity, and subunit composition [19]. Class I and II synthases are formed from one PhaC subunit, class III forms a heterodimer from PhaC–PhaE subunits, and class IV forms a heterodimer from PhaC–PhaR subunits [20]. Class I, III, and IV synthases favor short-chain monomers, thus preferentially producing scl PHAs. Interestingly, class II prefers medium length monomers, thus producing mcl PHAs [21].

The aim of this study was to identify PHA synthases and other genes involved in PHA production (*fabG*, *phaA*, *phaB*, *phaG*, and *phaJ*) in bacterial isolates from food to distinguish strains that can produce PHA from certain substrates. PHA production was verified using glucose, fructose, sunflower oil, and propionic acid as substrates.

## 2. Results

Sixty strains were obtained by isolation from fruits, vegetables, meat, and dairy products. These isolates were identified by MALDI-TOF or 16S rRNA sequencing (Appendix A). The identification revealed that a total of seven families were detected, of which there were fourteen strains of *Bacillaceae*, twenty-six strains of *Enterobacteriaceae*, six strains of *Moraxellaceae*, nine strains of *Pseudomonadaceae*, two strains of *Staphylococcaceae*, and three strains of *Xanthomonadaceae*. Screening of genes was performed with designed groups of primers in three different multiplex PCRs (2.2). *Cupriavidus necator* ATCC 17699, *Pseudomonas aeruginosa* ATCC 27853, and *Pseudomonas mendocina* ATCC 25411 were positive for some detected genes, and these strains were then used as controls (Table 1). According to Montenegro et al. [22], *Escherichia coli* DH5α was used as a negative control.

The results of the identification, genotypic, and phenotypic detection of PHA production are shown in Table 2, Table 3 and Appendix A. Twenty-six strains were classified into the family *Enterobacteriaceae* (genera *Escherichia*, *Klebsiella*, *Lelliottia*, *Pantoea*, and *Rahnella*) from food of plant origin. However, only *Escherichia coli* strains were found in foods of animal origin. Fourteen strains of the family *Bacillaceae* were assigned into genera *Bacillus*, *Lysinibacillus*, *Oceanobacillus*, *Peribacillus*, and *Priestia*. All six strains from the family *Moraxellaceae* belong to the *Acinetobacter calcoaceticus* species. They were isolated from lettuce, celery stalk, and white cabbage. Nine strains isolated from fruit and vegetables were classified into the family *Pseudomonadaceae* as genera *Pseudomonas*, e.g., *P. extremorientalis* and *P. oryzihabitans.* Two *Stenotrophomonas maltophilia* strains were isolated from white radish and dairy products and *S. rhizophila* was isolated from beetroot. Two strains were assigned into the family *Staphylococcaceae*, and taxons *Staphylococcus succinus* and *Mammaliicoccus sciuri* were identified from white cabbage and cucumber.

The most frequently found PHA synthase genes were of class I (*phaSyn1*), which was detected in 42% of the tested bacteria (Table 2). 

Of the other genes involved in PHA production, *phaA* was found in 45% of the monitored strains. This frequent occurrence indicates that the tested bacteria most often utilized the sugar substrates to form scl PHAs, which was subsequently confirmed by the phenotype tests. Conversely, class II of the PHA synthase gene (*phaSyn2*) and *phaG* were the least present; they were present only in 5% of strains. Class II of the PHA synthase gene was only found in the *Pseudomonadaceae* family; therefore, it is likely that strains of this family will be able to produce mcl PHA.

The utilization of substrates (glucose, fructose, propionic acid, and sunflower oil) for PHA production was tested by Sudan black staining, Nile blue staining, and FTIR detection. Figure 2 shows the results of the Sudan black staining, where the dye binds strongly to the PHA granules and should remain linked even after the subsequent ethanol wash. As a result, the colonies of producers are dark and non-producing colonies are decolored. The staining with Nile blue (Figure 2) is based on the same principle, but PHA production is determined under ultraviolet light, where PHA-producing colonies are fluorescent. As staining might be unreliable, PHA production was also detected by FTIR. Figure 2 shows the FTIR spectra from in situ PHA detection. In the spectra of PHA producers, the main absorption peak at 1735 cm^−1^ corresponds to a C=O group, and characteristic peaks in the range of 1200 to 900 cm^−1^ can be assigned to C-O-C. Due to the possible deficiencies of the phenotypic detection methods, strains were considered positive for PHA production by using a certain substrate only if it was confirmed by at least two methods. 

Table 3 shows the results from the phenotypic detection of PHA production in food isolates and collection strains. As can be seen, about 30% of the tested strains were able to utilize carbohydrates (fructose and glucose) for PHA production. Conversely, sunflower oil was utilized by the lowest number of tested strains (10%). Among representatives of the families *Pseudomonadaceae* and *Staphylococcaceae*, the ability to produce PHA was always detected with at least one from all tested substrates. However, only three strains from the *Pseudomonadaceae* family could utilize all four tested substrates. Among 32 representatives from the *Bacillaceae*, *Enterobacteriaceae*, *Moraxellaceae*, and *Xanthomonadaceae* families, the ability to use at least one of the tested substrates for PHA production was not detected. Despite this finding, some of them may be capable of production from other substrates that were not included in the tests in this study.

As can be seen in Table 2, genes of PHA synthase class I, III, and IV (*phaSyn1*, *phaSyn3*, and *phaSyn4*), *phaA*, *phaB*, and *fabG*, were detected in the *Bacillaceae* family. These results indicate the ability to produce scl PHAs from carbohydrates as well as from related carbon sources via β-oxidation. However, phenotypic detection (Table 3) revealed only the ability to utilize fructose, glucose, and propionic acid. More than one-third of *Bacillus* strains were able to produce PHA only from saccharides.

From the *Burkholderiaceae* family, only the collection strain *Cupriavidus necator* ATCC 17699 was investigated in this study. Class I PHA synthase genes, *phaA*, and *phaB* were detected in this strain (Table 2), indicating the ability to use carbohydrates, which was also proven phenotypically (Table 3). Unlike the minority strains from the *Burkholderiaceae* family, the strains from the *Enterobacteriaceae* family were the most numerous, i.e., 27 strains. Nevertheless, only nine strains could use one of the tested substrates (Table 3). All six representatives from the *Moraxellaceae* family were identified as *Acinetobacter calcoaceticus* (Appendix A). The probable ability to produce scl PHAs from related and unrelated carbon sources was genetically demonstrated for this species (Table 2). Phenotypic tests confirmed this suggestion; PHA production was also verified from all tested sources except for glucose (Table 3). All screened genes, except PHA synthase class IV, were found in the *Pseudomonadaceae* family (Table 2). Due to the presence of these genes, *Pseudomonadaceae* produced PHA from all tested carbon sources (Table 3). The most detected genes and the ability to produce PHA from all sources were proven in the *Pseudomonas extremorientalis* strain, which has tremendous potential for use in biotechnology. The production of PHA from each substrate was equally proven in the family *Staphylococcaceae* (Table 3), although only class I of PHA synthase and *phaJ* genes were detected (Table 2). Perhaps these products were produced by a different biosynthetic pathway than those observed. Unlike strains from the *Staphylococcaceae* family, members of the family *Xanthomonadaceae* could only utilize fructose for PHA production (Table 3).

When comparing the results of the molecular detection with the phenotypic detection in individual strains, PHA production was correctly predicted by PCR in almost 60% of strains. The prediction efficiency could be improved by increasing the number of tested substrates and monitoring more biosynthetic pathways.

## 3. Discussion

The molecular detection and subsequent phenotypic verification of the production of PHA can lead to the understanding of the biosynthetic pathways and thus to the prediction of the appropriate substrate and the resulting type of PHA formed. As a result, new producing strains may be discovered presenting a cheaper and more available substrate, leading to lower production costs of PHAs [10,30]. 

In this study, the genes of all four classes of PHA synthases: *phaC* (class I and II), *phaE* (class III), and *phaR* (class IV), which catalyze the polymerization of monomeric units, were screened [19]. Furthermore, screening of other genes involved in the biosynthesis of PHA was performed. Specifically, genes were involved in the production of scl PHAs from unrelated sources such as sugars (*phaA* and *phaB*) [16,17], in the production of PHAs from related sources by β-oxidation (*fabG* and *phaJ*) [14], and in the production of mcl PHAs from unrelated sources via the fatty acid synthesis pathway (*phaG*) [18]. Subsequently, the ability to produce PHA from glucose, fructose, sunflower oil, and propionic acid was monitored. PHA can be synthesized from propionic acid by the pathway from both related (*fabG* and *phaJ*) and unrelated (*phaA* and *phaB*) sources [31].

A total of 64 strains from seven families were investigated. A class IV PHA synthase gene was detected in the *Bacillaceae* family; as has also been shown in other studies [28,32]. The ability to use lignocellulosic biological waste and sugars to create homopolymers and copolymers was demonstrated [33] in *Bacillus* spp. This ability to create homopolymers and copolymers is probably due to the proven presence of *phaA*, *phaB*, and *fabG* genes and the utilization of both sugars and propionic acid. This result was confirmed by the research of Mohandas et al. [34], who demonstrated the ability to produce P(3HB-co-3HV) in *Bacillus cereus*. The resulting copolymer contained 12 mol.% of P(3HV) when using crude glycerol; however, propionic acid increased the production of P(3HV) by 30 mol.%. It seems that the synthesis of an application-specific polymer, including homopolymer P(3HB), can be achieved with a suitably chosen substrate composition. The production of P(3HB) homopolymer by *Bacillus thuringiensis* and *Bacillus cereus* from glucose was demonstrated in a study by Ray and Kalia [35]. The ability to use inexpensive substrates from waste transformer oil, waste-derived volatile fatty acids, and other types of waste were reported [30,36]. *Priestia megaterium* was identified among the isolates which produced PHA from sugars and propionic acid. This interesting strain is useful for producing small molecules such as vitamin B12, polymers such as P(3HB), and even protein, and is suitable for biotechnological applications [37]. A major advantage of the *Bacillaceae* family is the absence of endotoxins in the outer membrane and that PHAs obtained from them are suitable for medical applications [38]. 

As with the *Bacillaceae* family, a representative of the *Burkholderiaceae* family (*Cupriavidus neactor* ATCC 17699) was demonstrated to utilize fructose, glucose, and propionic acid. *Cupriavidus neactor* is one of the most studied PHA producers. Previous studies have demonstrated the presence of all found genes (PHA synthase class I, *phaA*, and *phaB*) in this bacterium [39]. Various substrates such as food waste-derived volatile fatty acids, polyethylene from waste Tetra Pak packaging, and starchy waste were used in the low-cost PHA production [40,41].

The largest tested family was *Enterobacteriaceae*, in which genes involved in PHA production were detected in almost 90% of the strains; however, only over 30% were phenotypically proven to produce PHA. *Escherichia coli* is considered a non-PHA-producing bacterium; therefore, it is used to create recombinant PHA-producing strains [42,43]. However, PHA production was detected in the species involved in this study. Although recent papers have not demonstrated the production ability of *Escherichia coli*, some wild strains may produce PHA in response to stressful conditions. In further studies, it would be appropriate to focus on these strains, verify the production by other methods, and characterize the eventual products. None of the searched genes were detected in *Klebsiella oxytoca*; however, the ability to utilize propionic acid was demonstrated. Previous works showed the ability to produce PHA from xylose in the genus *Klebsiella* [44]. Furthermore, members of the genus *Klebsiella* produced P(3HB-co-3HV) from hardwood sulfite [45]. Consequently, the genus *Klebsiella* seems to produce PHA by a different pathway than those described in this study. Other genera from this family were *Pantoea* and *Rahnella*, which could not use the tested substrates. Nevertheless, this genus ranks among the documented PHA producers [46,47]. Therefore, the production is probably strain-dependent. On the other hand, *Lelliottia amnigena* produced PHA from propionic acid; however, this strain has not been investigated in earlier studies.

In *Acinetobacter calcoaceticus*, the genes that predicted the ability to synthesize PHA from related and unrelated sources were found, and production from fructose, sunflower oil, and propionic acid was confirmed. The production of scl and mcl PHAs from oil and sugars has previously been detected in *Acinetobacter* sp. [48]. Most of the monitored genes were found in the *Pseudomonadaceae* family. The members of this family utilize a variety of substrates [49]. Class II of PHA synthase, detected in several strains, is typical of the genus *Pseudomonas* and can generate mcl PHAs [50]. For example, Rai et al. [51] produced poly(3-hydroxyoctanoate) (P(3HO)) from sodium octanoate using *Pseudomonas mendocina*. The members of this family can advantageously use waste carbohydrates and oils, which can lead to a reduction in PHA production costs [52]; additionally, they can even utilize pollutants (e.g., phenol) to produce PHA [53].

The *Staphylococcaceae* family produced PHA from all substrates tested. Utilization of diverse substrates was found by Wong et al. [9], in which *Staphylococcus epidermis* produced P(3HB) from malt, milk, sesame oil, soybean waste, and vinegar. Class I of the PHA synthase gene demonstrated in this study may allow them to make this product. The last family included in this study was *Xanthomonadaceae*, in which genes involved in PHA production from both carbohydrates and fats were revealed. However, the PHA production was phenotypically observed only from fructose. Nevertheless, the ability to produce PHA from glucose, wood chips, cardboard cutouts, plastic bottle cutouts, shredded polystyrene cups, plastic bags, and potato starch has been described in the literature [54]. Therefore, it is possible that the production is again strain dependent.

As PHA production from diverse substrates was detected in this study and their biosynthetic pathways were revealed, future experiments would be appropriate to characterize these products and monitor the effect of substrate combinations on the resulting mechanical properties and thereby establish conditions for application-specific PHA production. Additionally, the utilization of waste sources by proven producers could be tested, which would reduce production costs.

## 4. Materials and Methods 

### 4.1. Materials and Chemicals

The collection strains—*Cupriavidus necator* ATCC 17699, *Pseudomonas aeruginosa* ATCC 27853, and *Pseudomonas mendocina* ATCC 25411—were provided by the Czech Collection of Microorganisms (CCM, Brno, Czech Republic). *Escherichia coli* DH5α was obtained from Takara Bio Europe SAS, Paris, France.

*Stenotrophomonas maltophilia* strain was isolated from a dairy product and provided by the Dairy Research Institute in Prague (Prague, Czech Republic). The other 59 bacterial strains were isolated from fruits, vegetables, and meat bought at the local market in the Czech Republic (Appendix A). Ten grams of food sample was homogenized in 90 mL of sterile saline solution and spread on the plates with the selective media: Baird-Parker Agar, Endo Agar, Pseudomonas Selective Agar (Sigma-Aldrich, St. Louis, MO, USA), Clostridium Agar, M17 Agar, Mannitol Salt Agar, Violet Red Bile Agar (HiMedia Laboratories GmbH, Mumbai, Maharashtra, India), and MRS agar (Oxoid, Basingstoke, UK) and incubated at 30 or 37 °C for 24 h. 

Selected isolates were identified by Microflex MALDI-TOF MS mass spectrophotometery (Bruker Daltonics, Bremen, Germany). Samples for MALDI-TOF identification were prepared by mixing the bacterial culture with 150 μL of sterile distilled water and 450 μL of 96% ethanol (Lach-Ner, Neratovice, Czech Republic). Then, the samples were centrifuged for 2 min at 14,000 rpm. After centrifugation, the supernatant was separated, and the pellets were re-centrifuged. The remains of the supernatant were removed and the pellets were dried. The pellets were mixed with 10 μL of 70% formic acid (Merck KGaA, Darmstadt, Germany) and 10 μL of acetonitrile (Sigma-Aldrich, St. Louis, MO, USA). The suspensions were centrifuged at 14,000 rpm for 2 min and 1 μL of the supernatants was applied to the MALDI plate. Following drying, every sample was overlaid with 1 μL of HCCA (2-Cyano-3-(4-hydroxyphenyl) acrylic acid) (Bruker Daltonics, Bremen, Germany) matrix and dried again. The resulting samples were ionized with a nitrogen laser (wavelength of 337 nm and frequency of 20 Hz). The results were evaluated by the MALDI Biotyper 3.0 identification database (Bruker Daltonics, Billerica, MD, USA) [55]. Bacterial strains with a score value lower than 2.000 were subjected to 16S rRNA sequencing. 

Strains were cultivated in brain heart infusion agar or M17 agar media (both HiMedia Laboratories GmbH, Mumbai, Maharashtra, India). Mineral agar media containing 20 g/L of carbon (glucose, fructose, propionic acid, or sunflower oil (Sigma-Aldrich, St. Louis, MO, USA)) was used for phenotypic screening. The composition of the mineral agar media was 3 g (NH_4_)_2_SO_4_, 11.1 g Na_2_HPO_4_·12H_2_O, 1.05 g KH_2_PO_4_, 0.2 g MgSO_4_·7H_2_O, 1 mL solution of trace elements, and 15 g agar (Sigma-Aldrich, St. Louis, MO, USA) per liter of distilled water. One liter of solution of trace elements contained 9.7 g FeCl_3_·6H_2_O, 7.8 g CaCl_2_·2H_2_O, 0.156 g CuSO_4_·5H_2_O, 0.119 g CoCl_2_·6H_2_O, 0.118 g NiCl_2_·6H_2_O, and 0.1 g ZnSO_4_·7H_2_O in 0.1 M HCl. Tween 80 (Sigma-Aldrich, St. Louis, MO, USA) (5 g/L) was added into a mineral agar media with sunflower oil as an emulsifier. Levels of the mineral agar media pH were adjusted to 7.

Primers were designed to detect genes involved in PHA production using Primer3 (v. 0.4.0, Singapore). Gene sequences were obtained from the European Nucleotide Archive (for genes *phaJ* and *phaG*) and the National Center for Biotechnology Information (for the remaining primers) were used to design the primers. In addition, fabGF and fabGR primers were used to detect the *fabG* gene [25]. All primers and the size of PCR products are listed in Table 4.

### 4.2. Molecular Detection

Isolation of DNA was performed using a NucleoSpin Tissue kit (Macherey-Nagel, Düren, Germany). The purity and concentration of the isolated DNA were verified spectrophotometrically at 260 and 280 nm by a Tecan Infinite^®^ 200 PRO (Tecan, Männedorf, Switzerland). After isolation, 1 µL (for the detection of one or two genes) or 2 µL (for the detection of three or more genes) of DNA was mixed with 10 µL GoTaq^®^ Hot Start Green Master Mix (Promega, Madison, WI, USA) and 25 µM of each primer (Eastport–Metabion, Prague, the Czech Republic). The PCR reaction was performed in Aeris™ thermocycler (ESCO, Singapore). The PCR program is shown in Table 5.

An optimal annealing temperature of 65 °C was determined by gradient PCR, and suitable groups of primers were created: four pairs of primers detecting PHA synthases (*phaSyn1*, *phaSyn2*, *phaSyn3*, and *phaSyn4*), three pairs of primers for *phaB*, *phaG*, and *phaJ* genes and two pairs of primers for *fabG* and *phaA* genes. Subsequent gene screening of bacterial isolates was carried out using this optimized multiplex PCR method.

Following the PCR reaction, PCR products were resolved by agarose gel electrophoresis on 1.5% agarose gel (Lonza, Rockland, ME, USA) and visualized with GelRed™ (Biotium, Fremont, CA, USA; 10 µL/100 mL of gel). PeqGOLD 100 bp Plus (VWR Peqlab, Erlangen, Germany) was used as a DNA marker.

### 4.3. Phenotypic Detection

#### 4.3.1. Nile Blue Staining

The tested bacteria were cultivated in mineral agar media containing 20 g/L of the carbon source (glucose, fructose, propionic acid, or sunflower oil) for 72 h. Then, the bacterial colonies were stained with 0.05% Nile blue (Sigma-Aldrich, St. Louis, MO, USA) solution in ethanol (Lach-Ner, Neratovice, Czech Republic) for 20 min in the dark. After staining, detection was performed under UV light, where cells accumulating PHA exhibited fluorescence [56,57].

#### 4.3.2. Sudan Black Staining

After 72 h of bacterial cultivation in mineral agar media containing 20 g/L of the carbon source (glucose, fructose, propionic acid, or sunflower oil), the bacterial colonies were stained with 0.02% Sudan black solution in ethylene glycol (Sigma-Aldrich, St. Louis, MO, USA) for 30 min. Then, stained colonies were washed with 96% ethanol (Lach-Ner, Neratovice, Czech Republic). PHA producers were identified according to dark colonies and non-producing bacterial strains were decolored [58,59].

#### 4.3.3. Fourier Transform Infrared Spectroscopy

Detection of PHA production by Fourier transform infrared (FTIR) spectroscopy was performed according to Arcos-Hernandez et al. [60]. After 72 h of bacterial cultivation in mineral agar media containing 20 g/L of the carbon source (glucose, fructose, propionic acid, or sunflower oil), the bacterial colonies were collected, suspended in 1 mL of isotonic saline (9 g/L NaCl (Sigma-Aldrich, St. Louis, MO, USA)), and centrifuged at 15,000× *g* for 5 min. After centrifugation, the supernatant was separated and the bacterial pellets were washed with 90 µL of isotonic saline and re-centrifuged. The supernatant was again separated and the bacterial pellets were resuspended in 80 µL of isotonic saline. After that, the bacterial suspension was centrifuged once more, and a small amount of wet bacterial pellet was collected using an inoculation loop and spread into a thin layer on a microscopic glass slide. The slides were dried at 105 °C for 5 min and then Fourier transform infrared spectroscopy (FTIR) analysis was performed on Nicolet iS10 (Thermo Fisher Scientific, Waltham, MA, USA) in ATR mode fitted with a diamond crystal. The data were evaluated with OMNIC 8 software (Thermo Fisher Scientific, Waltham, MA, USA). Measurement conditions were 32 scans at the resolution of 4 cm^−1^ and in the range of 4000 to 500 cm^−1^.

## 5. Conclusions

In this study, genes of all four classes of PHA synthase and other genes involved in PHA production (*fabG*, *phaA*, *phaB*, *phaG*, and *phaJ*) were screened in 64 bacterial collection strains and food isolates. Subsequently, the ability to produce PHA from fructose, glucose, sunflower oil, and propionic acid was tested using Sudan black staining, Nile blue staining, and FTIR detection. Class I of PHA synthase and *phaA* were detected most frequently (in more than 40% of strains), which suggests the ability to produce scl PHAs from unrelated sources, such as carbohydrates. Phenotypically, this ability was confirmed in around 30% of the samples. The most diverse genes were found in the genus *Pseudomonas*. *Pseudomonas extremorientalis* is the most promising PHA producer among all tested strains from all tested carbon sources and is therefore suitable for further study to reveal its biotechnological potential. 

## Figures and Tables

**Figure 2 ijms-24-01250-f002:**
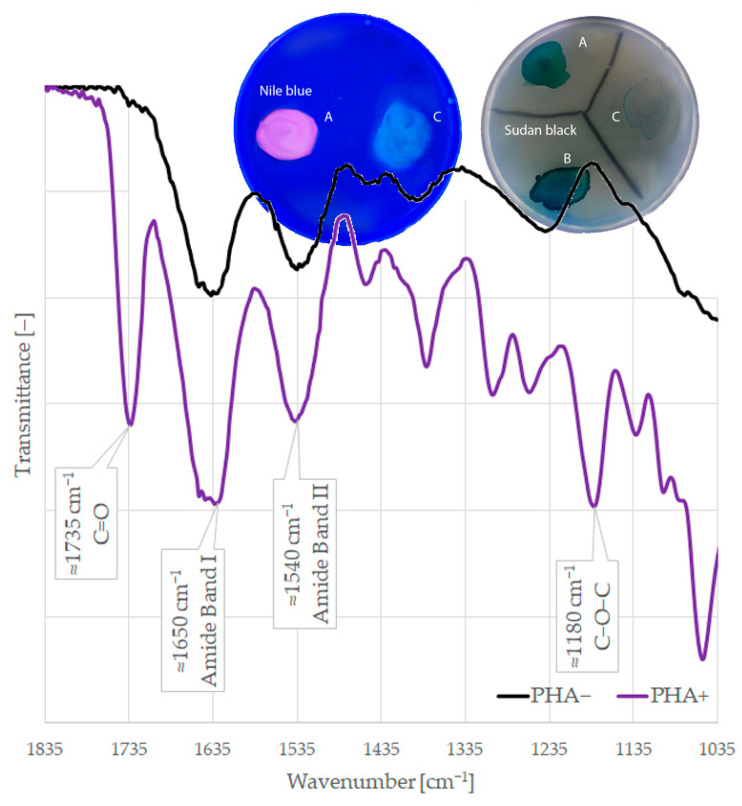
Phenotypic detection of PHA production: Nile blue staining; Sudan black staining; and detection by FTIR (A) *Cupriavidus neactor* ATCC 17699, (B) *Pseudomonas extremorientalis* are PHA+ and (C) *Escherichia coli* DH5α is PHA−.

**Table 1 ijms-24-01250-t001:** The list of strains (collection and food isolates) with detected genes used as positive controls for multiplex PCRs.

Strain	Genes	Related References
*Cupriavidus necator* ATCC 17699	*phaA*, *phaB*, *phaC* (class I)	[23]
*Pseudomonas mendocina* ATCC 25411	*phaC* (class II)	[24]
*Pseudomonas aeruginosa* ATCC 27853	*phaJ*, *fabG*	[25,26]
*Stenotrophomonas maltophilia*	*phaE* (class III)	[27]
*Priestia megaterium*	*phaR* (class IV)	[28]
*Pseudomonas putida*	*phaG*	[29]

**Table 2 ijms-24-01250-t002:** Genotypic detection of four classes of PHA synthases and other genes involved in PHA formation (%) in food isolates and collection strains from seven families.

% *	*n*	*phaSyn1*	*phaSyn2*	*phaSyn3*	*phaSyn4*	*phaA*	*phaB*	*phaG*	*fabG*	*phaJ*
Total	64	42.2	4.7	10.9	15.6	45.3	18.8	4.7	12.5	10.9
*Enterobacteriaceae*	27	51.9	0.0	0.0	18.5	74.1	11.1	3.7	3.7	3.7
*Bacillaceae*	14	14.3	0.0	7.1	28.6	21.4	14.3	0.0	7.1	0.0
*Pseudomonadaceae*	11	45.5	27.3	27.3	0.0	36.4	27.3	18.2	36.4	27.3
*Moraxellaceae*	6	33.3	0.0	16.7	0.0	16.7	33.3	0.0	16.7	16.7
*Xanthomonadaceae*	3	33.3	0.0	66.7	33.3	0.0	33.3	0.0	33.3	33.3
*Staphylococcaceae*	2	100.0	0.0	0.0	0.0	0.0	0.0	0.0	0.0	50.0
*Burkholderiaceae*	1	100.0	0.0	0.0	0.0	100.0	100.0	0.0	0.0	0.0

* Percentage of positive strains from each family.

**Table 3 ijms-24-01250-t003:** Phenotypic detection of PHA production (%) from four different carbon sources (feedstock) in food isolates and collection strains from seven families.

% *	*n*	Feedstock			
		Fructose	Glucose	Sunflower Oil	Propionic Acid
Total	64	29.7	28.1	9.4	26.6
*Enterobacteriaceae*	27	3.7	3.7	0.0	22.2
*Bacillaceae*	14	42.9	35.7	0.0	28.6
*Pseudomonadaceae*	11	72.7	90.9	36.4	36.4
*Moraxellaceae*	6	16.7	0.0	16.7	16.7
*Xanthomonadaceae*	3	33.3	0.0	0.0	0.0
*Staphylococcaceae*	2	50.0	50.0	50.0	50.0
*Burkholderiaceae*	1	100.0	100.0	0.0	100.0

* Percentage of positive strains from each family.

**Table 4 ijms-24-01250-t004:** Used primers and sizes of their PCR products.

Primer	Gene	Primer Sequence (5′–3′)	Product Size (bp)	Ref.
phaAF	*phaA*	CCATGACCATCAACAAGGTG	262	This study
phaAR		TATTCCTTGGCCACGTTCTC		
phaBF	*phaB*	AATGTGGCTGACTGGGACTC	164	This study
phaBR		GAGGTCAGGTTGGTGTCGAT		
phaSyn1F	*phaC* (Class I)	TGCGCAACATGATGGAAGAC	204	This study
phaSyn1R		AGTACTTGTTGATGCACGGC		
phaSyn2F	*phaC* (Class II)	TACATCGAGGCGCTCAAGGA	594	This study
phaSyn2R		GCCGAACAGATGAGCCGATT		
phaSyn3F	*phaE* (Class III)	CAGTGGATGCTGCAGGGC	463	This study
phaSyn3R		GCAGGCCGATACGTTCACTC		
phaSyn4F	*phaR* (Class IV)	AAATGAAGTAACGGGGCGCT	326	This study
phaSyn4R		CCTGAAGCTGCTCACCTTGA		
phaJF	*phaJ*	CGAGTACACCAGCAGCATCG	287	This study
phaJR		GGTTCTTCGGCAGCTTCTCG		
fabGF	*fabG*	TGGCTCGAGAGAGAGAAAGGAGA	750	[25]
fabGR		TTCCGCAACGAATTCTAGAC		
phaGF	*phaG*	AGAAGGCAGTGGTGAGTTCG	425	This study
phaGR		ACAGGCGGTCTTGTTTTCCA		

**Table 5 ijms-24-01250-t005:** The PCR program.

Step Name	Temperature (°C)	Time (min)	Number of Cycles
Initial denaturation	95	5	1
Denaturation	95	0.5	40
Annealing	65	0.5	
Elongation	72	1	
Final elongation	72	10	1
Cooling	4	∞	1

## Data Availability

Not applicable.

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
