# Peer review of "Genotypic and Phenotypic Detection of Polyhydroxyalkanoate Production in Bacterial Isolates from Food"

_ijms, 2023, doi:10.3390/ijms24021250_

Round 1

Reviewer 1 Report

The paper “Genotypic and Phenotypic Detection of Polyhydroxyalkanoate Production in Bacterial Isolates from Food” is devoted to studying of different methods which can be used to identify PHA bacterial producers. The data obtained in the paper are interesting and suitable to publish in International Journal of Molecular Sciences. However, it needs to address some comments, and thus require substantial major revision to improve the quality of the manuscript.

1. The aim of the work is to distinguish strains that can produce PHA of specific properties. Properties are known to be determined by the composition of PHA. Did you determine the composition of the polymers synthesized by the studied strains?

2. Why did you use IR rather than gas chromatography to identify PHA?

3. Check all generic and specific names of microorganisms (especially in References). They must be in Italics.

4. Table 5. Describe the data in the table in more detail and clearly.

5. Add information about MALDI-TOF to Materials and Methods section.

6. Volatile acids including propionic acid are known to be toxic to bacteria. You used propionic acid at a concentration of 20 g/L. Was this concentration not toxic to the bacteria?

7. You write that Pseudomonas extremorientalis strain was determined as prospective for biotechnology applications. Did you evaluate biomass yield and polymer content as well as its composition?

Author Response

Dear Editor and Reviewers, 

Thank you for the opportunity to submit a revised draft of our manuscript entitled Genotypic and Phenotypic Detection of Polyhydroxyalkanoate Production in Bacterial Isolates from Food to the Special Issue “SMART and Macromolecular Biomaterials, from Materials to Biology”. We appreciate the time and effort that you and the reviewers have dedicated to providing your valuable feedback on our manuscript. We are grateful to the reviewers for their insightful comments on our paper. We have been able to incorporate changes to reflect all the suggestions provided by the reviewers. We have highlighted the changes in revision mode.

Sincerely, 

Pavel Pleva et al. 

(corresponding author) 

Tomas Bata University in Zlin

Faculty of Technology 

Department of Environmental and Protection Engineering 

Vavreckova 275 

760 01 Zlin

Czech Republic 

Response to the Reviewer 1

Comments and Suggestions for Authors

The paper “Genotypic and Phenotypic Detection of Polyhydroxyalkanoate Production in Bacterial Isolates from Food” is devoted to studying of different methods which can be used to identify PHA bacterial producers. The data obtained in the paper are interesting and suitable to publish in International Journal of Molecular Sciences. However, it needs to address some comments, and thus require substantial major revision to improve the quality of the manuscript.

  1. The aim of the work is to distinguish strains that can produce PHA of specific properties. Properties are known to be determined by the composition of PHA. Did you determine the composition of the polymers synthesized by the studied strains?

Response: There is no reliable possibility to determine monomer composition of the PHB other than hydrolysis and chromatographical separation of the purified material. But even this would not be sufficient to characterize the real properties. Mechanical and thermal tests would be necessary for this. A relatively high amount of the material and work would be thus needed to characterize each strain. Our study was aimed to screen a maximum number of strains and use genetical markers to select the interesting strains. To address your concerns, we modified the aims in the manuscript.

  1. Why did you use IR rather than gas chromatography to identify PHA?

Response: As noted above, chromatography needs a higher amount of purified polymer, also even such characterization would not be sufficient to obtain relevant mechanical and processing characteristics of the material. In this study, we focused on finding and applying the methods that enable to screen large number of strains and identify the perspective ones.

  1. Check all generic and specific names of microorganisms (especially in References). They must be in Italics.

Response: Thank you for the note. Names have been checked and corrected.

  1. Table 5. Describe the data in the table in more detail and clearly.

Response: Thank you for the comment, Table 5. has been described in more detail and clearly.

  1. Add information about MALDI-TOF to the Materials and Methods section.

Response: Thank you, we have added the description of the method.

  1. Volatile acids including propionic acid are known to be toxic to bacteria. You used propionic acid at a concentration of 20 g/L. Was this concentration not toxic to the bacteria?

Response: Yes, you are right. Propionic acid can be toxic to some bacteria. The aim of this experiment was to select strains that are able to utilize propionic acid as a carbon source, thus they are able to grow.

  1. You write that Pseudomonas extremorientalis strain was determined as prospective for biotechnology applications. Did you evaluate biomass yield and polymer content as well as its composition?

Response: In this study and at this stage of research we identified the strain as promising based on the ability to utilize all tested substrates and the presence of important genes. Pseudomonas extremorientalis is the most promising PHA producer among all tested strains and therefore it is suitable for further study to reveal its biotechnological potential. We agree that the biomass and polymer yield are important properties for the further study and application of the strain.

Reviewer 2 Report

I have to reject this paper due to low merit. Manuscript just reports PHA production from ATCC strains. There is nothing novel in it and no application has been associated with produced PHA. At least 1-2 experimental assays, demonstrating worth of prodcued PHA should have been included. Keeping in view high calibre of science published in IJMS, this paper should be redirected elsewhere.

Author Response

Dear Editor and Reviewers, 

Thank you for the opportunity to submit a revised draft of our manuscript entitled Genotypic and Phenotypic Detection of Polyhydroxyalkanoate Production in Bacterial Isolates from Food to the Special Issue “SMART and Macromolecular Biomaterials, from Materials to Biology”. We appreciate the time and effort that you and the reviewers have dedicated to providing your valuable feedback on our manuscript. We are grateful to the reviewers for their insightful comments on our paper. We have been able to incorporate changes to reflect all the suggestions provided by the reviewers. We have highlighted the changes in revision mode.

Sincerely, 

Pavel Pleva et al. 

(corresponding author) 

Tomas Bata University in Zlin

Faculty of Technology 

Department of Environmental and Protection Engineering 

Vavreckova 275 

760 01 Zlin

Czech Republic 

Response to the Review 2

Reviewer comments  

I have to reject this paper due to low merit. Manuscript just reports PHA production from ATCC strains. There is nothing novel in it and no application has been associated with produced PHA. At least 1-2 experimental assays, demonstrating worth of prodcued PHA should have been included. Keeping in view high calibre of science published in IJMS, this paper should be redirected elsewhere.

Response: Thank you very much for your deep insight. The aim of this study was to screen the ability of PHA production among food isolates. We want to emphasize that the manuscript describes an extensive screening resulting into 60 newly characterized PHA producing isolates.

Along with this isolation effort we also worked with a few ATCC collection strains which were used here as a reference for the verification of methods. We tried to modify the manuscript to better highlight all aspects of our study.

Let us kindly summarize a brief recapitulation of the changes made specifically in the response to your objection that follows:

Abstract

16 -  In the study, all four classes of PHA synthases and other genes involved in PHA formation (fabG, phaA, phaB, phaG, and phaJ) were detected by PCR in 64 bacterial collection strains and food isolates.

  1. Introduction

98 - The aim of this study was to identify PHA synthases and other genes involved in PHA production (fabG, phaA, phaB, phaG, and phaJ) in bacterial isolates from food to distinguish strains that can produce PHA of specific properties.

  1. Materials and Methods

2.1. Materials and Chemicals

107 - Stenotrophomonas maltophilia strain was isolated from a dairy product and provided by the Dairy Research Institute in Prague (Prague, Czech Republic). Other 59 bacterial strains were isolated from fruits, vegetables, and meat bought in the local market in the Czech Republic (Table S1).

  1. Results

194 - Sixty strains were obtained by isolation from fruits, vegetables, meat, and dairy products.

205 – 217 The results of the identification, genotypic and phenotypic detection of the PHA production are shown in Tables (Table 4, 5, S1). Twenty six strains were classified into the family Enterobacteriaceae (genera Escherichia, Klebsiella, Lelliottia, Pantoea, Rahnella) from food of plant origin. However, only Escherichia coli strains were found in food of animal origin. Fourteen strains of the family Bacillaceae were assigned into genera Bacillus, Lysinibacillus, Oceanobacillus, Peribacillus, Priestia. All six strains from the family Moraxellaceae belong to the Acinetobacter calcoaceticus species. They were isolated from lettuce, celery stalk, and white cabbage. Nine strains isolated from fruit and vegetables were classified into the family Pseudomonadaceae as genera Pseudomonas, e.g. P. extremorientalis, P. oryzihabitans. Two Stenotrophomonas maltophilia strains where isolated from white radish and dairy product and S. rhizophila was isolated from beetroot. Two strains were assigned into the family Staphylococcaceae, and taxons Staphylococcus succinus and Mammaliicoccus sciuri were identified from white cabbage and cucumber.

  1. Conclusions

381- In this study, genes of all 4 classes of PHA synthase and other genes involved in PHA production (fabG, phaA, phaB, phaG, and phaJ) were screened in 64 bacterial collection strains and food isolates.

In this study and at this stage of research we identified the strain as promising based on the ability to utilize all tested substrates and the presence of important genes. Pseudomonas extremorientalis is the most promising PHA producer among all tested strains and therefore it is suitable for further study to reveal its biotechnological potential. We agree that the biomass and polymer yield are important properties for the further study and application of the strain.

Round 2

Reviewer 2 Report

Since the authors have demonstrated novelty by targeting four classes of enzymes and some analysis with ATCC strains as well, I recommend acceptance.